# OpenReview forum: "Efficient Adversarial Training in LLMs with Continuous Attacks"
_NeurIPS.cc/2024/Conference — NeurIPS 2024 spotlight_

### Official Review · Reviewer_Z2aE · 2024-06-21

**Soundness:** 2
**Presentation:** 3
**Contribution:** 3
**Rating:** 5
**Confidence:** 3

**Summary:**

Large language models (LLMs) are vulnerable to adversarial attacks that can bypass their safety guardrails. However, current adversarial training methods for LLMs are hindered by the high computational costs required to perform discrete adversarial attacks at each training iteration. To solve this problem, the paper instead calculates adversarial attacks in the continuous embedding space of the LLM, which is orders of magnitudes more efficient.

**Strengths:**

The presentation of the article is clear.

The proposed method can consume less computational cost to obtain robust LLMs against adversarial attacks.

**Weaknesses:**

1. The main claim of the paper is that the proposed method can use less computational cost to obtain a robust LLM, and regarding this viewpoint, there is only an experimental setting presented in Table 1, which may be insufficient. I believe that some more intuitive experiments should be added, such as comparative experiments on the real-time cost for a single example and for the entire process.

2. Although the paper is overall clear and the problem it intends to solve is well-defined, some details are not clear enough, especially the description of the proposed method's pipeline. For example, 'IPO' keeps appearing in the first two pages, but only in Section 3 are some descriptions of 'IPO' provided.

3. The hardware description in Section 4 is also unclear. I understand that this does not impact the main contributions of the paper, but a more explicit explanation of each experiment's settings would provide a clearer understanding for the readers. Additionally, totaling GPU hours across different GPUs may be unreasonable, as 1904 GPU hours on a V100 differ from those on 80GB A100 GPUs. Therefore, I hope the authors can further refine these details.

4. As LLAMA-2 is one of the most mainstream open-source LLMs, why have the experiments yet to be conducted on LLAMA-2 to evaluate the effectiveness of the proposed method?

5. Currently, the defense mechanisms on LLMs are not limited to R2D2 nor even to AT methods. I'm curious about how the proposed method compares with other methods, such as some methods mentioned in the related works.

**Questions:**

See Weaknesses.

**Limitations:**

The authors have discussed limitations of the work.

---

> ### Author Rebuttal · Authors · 2024-08-05
>
> We thank the reviewer for their insightful comments and address each point/question in turn:
>
> **W1: The authors should conduct a controlled experiment to measure runtime differences between the approaches:**
>
> **A1:** This is a valid concern and we address it by measuring the wall time it takes to run $15$ steps of the algorithms in both cases, extrapolating from there also the total experiment time, which would be prohibitive to run for us. The wall times are in the ballpark as our forward pass comparison:
>
> On an A100 a single step of the R2D2 took the 489.7 longer than a single step of C-AdvIPO. Extrapolating this results to the total number of step considered for training R2D2, its complete training would have taken 1991 times longer with the implementation provided in the repository.
>
> **W2: Important notation and methods are not always introduced at the right point in the paper:**
>
> **A2:** Thank you for pointing this out! We will improve the writing further, for instance by better defining the term IPO earlier in the manuscript and in more detail.
>
> **W3: The hardware descriptions are imprecise**:
>
> **A3:** Roughly 800 hours were on 80GB A100, the remainder were on the 40GB GPUs. Only 11.6 hours were used on V100 GPUs, which were only used for debugging.
>
> **W4: Llama2 should be added as a baseline:**
>
> **A4:** We thank the reviewer for the suggestion as Llama2 is a popular model. We have trained Llama2 with the CAT loss successfully, which considerably improves the robustness of the model with minor degradations in utility. See the overall response and Figure 2 of the attached PDF for the results.
>
> **W5: More defenses should be included in the evaluation**:
>
> **A5:** We’d be happy to include any defenses the reviewer finds relevant. Could you point us to which defenses you would like to see?
>
> In the meantime, let us discuss some that come to mind to us and why we didn’t include them: Many of the other defenses mentioned are orthogonal to our paper, such as perplexity filters on the input or toxicity filters on the output. Other papers such as Rainbow Teaming are both orthogonal and not reproducible as the authors do not share code/data/models necessary to do so (Rainbow teaming for instance use a Meta-internal helpful-only Llama model). Other works only apply to Bert style models on don’t focus on generation. We also want to note that it is standard practice to compare adversarial training approaches with each other as the vast majority of empirical approaches have been broken by third-party evaluations, and adversarial training remains one of the only exceptions [4].
>
> (4) Tramer et al., “On Adaptive Attacks to Adversarial Example Defenses" NeurIPS, 2020

---

> > ### Comment · Reviewer_Z2aE · 2024-08-11
> >
> > Thanks for the responses. Most of my concerns have been addressed.
> >
> > However, I still have concerns about W5: The authors shared a series of similar works in the related work section, but failed to compare them in the experimental section. This can easily lead readers to question the accuracy of the limitations of related works.
> >
> > Of course, I agree with authors’ explanation that some methods are orthogonal or irreproducible. Nevertheless, this does not substantially solve the above problem, although I acknowledge your explanation. I hope the authors can handle this issue more appropriately in the future. Hence, I maintain my score.

---

> > > ### Author Response · Authors · 2024-08-11
> > >
> > > Thank you for your feedback on our paper. We'd like to clarify our position:
> > > The primary contribution of our paper is to address a specific research question:
> > > - Does adversarial training with continuous attacks in the token embedding space of an LLM provide robustness to discrete natural language attacks?
> > >
> > > Prior works on adversarial training in NLP target different objectives, such as improved generalization or adversarial robustness in sentiment classification. These methods differ in several key aspects:
> > > Previous work didn't consider threat models, where the attacker has complete input control but generally enforced similarity to the initial text. This was done to keep the adversarial nature of the perturbation (to be semantically not meaningful to a human observer). However, we focus on alignment and not classification, which results in substantially different threat models.
> > > Further, not all methods directly apply to decoder-only LLMs, and some only apply to pretraining, demanding unrealistic amounts of compute in the LLM setting.
> > >
> > > We want to emphasize that the scope of our paper is focused first on answering our research question and second on adversarial robust alignment algorithms. We believe the prior work mentioned above cannot be considered as a baseline that should be compared to our algorithms, as extending such prior work to our setting would be a contribution by itself. Instead, we adopted two of the most common losses used in LLM training. However, we acknowledge that leveraging such prior work to improve robustness of alignment is an interesting avenue for future work, and we will discuss this option in the outlook section of the final manuscript.

---

### Official Review · Reviewer_TJPv · 2024-07-07

**Soundness:** 2
**Presentation:** 2
**Contribution:** 3
**Rating:** 5
**Confidence:** 3

**Summary:**

This paper introduces adversarial attacks in continuous space in the context of LLMs. In addition, it utilizes continuous attacks for adversarial training to robustify LLMs and demonstrate that this efficient training algorithm can indeed protect LLMs against various attacks, including discrete ones, while maintaining the utility.

**Strengths:**

1. The proposed method in this paper is easy to understand, to implementation and generally applicable.

2. The proposed method dramatically improve the efficiency of adversarial training in the context of LLMs.

3. The proposed method does not hurt the utility of the model too much.

**Weaknesses:**

1. The experiments should be more comprehensive: (1) In Figure 2, the comparison between R2D2 and the proposed methods is only conducted on the model ZEPHYR-7B, comparisons on more models would be preferred. (2) The authors only test one safe answer ''Sorry, I can't do that''. The safe answer $y$ is a very important variable in the loss objective function. Tests on more ''safe answers'' would make the results more convincing and more generally applicable. (3) The attack success rates are based on GCG, AutoDAN and PAIR. Recently, there have been some stronger attacks proposed, such as LLM-adaptive attacks (https://github.com/tml-epfl/llm-adaptive-attacks), the authors should include more attacks for a more comprehensive evaluation. (4) Ablation studies should be conducted to support why IPO is better than DPO, as line 161 to 162 claimed.

2. The proposed methods introduce more hyper-parameters to tune, hindering its application for practitioners.

3. The presentation of the paper can be improved, for example, the formulation in line 154 to 156 is very confusing, as $\mathcal{L}'$ is not defined.

**Questions:**

The major concerns have been pointed out in the weakness part, the authors should first answer questions in that section. In addition, we have some minor questions:

1. In line 210 to 211, why do bigger models have smaller values of $\epsilon$? Is there any intuition?

2. Similar to TRADES [Zhang et. al 2019], in addition to the value of $\epsilon$, is it possible to add a co-efficient to the last term of Equation (4) to balance the trade-offs between robustness and utility?

Overall, the paper tackles an interesting problem. However, due to the weakness part and the questions above, I believe the manuscript needs to be edited to improve. I welcome the authors to address my concerns during the rebuttal and will re-evaluate the manuscript after the rebuttal.

> Post Rebuttal

I improved my ratings to 5 after reading authors' rebuttal.

**Limitations:**

The limitations and the societal impact are discussed at the end of the paper.

---

> ### Author Rebuttal · Authors · 2024-08-05
>
> We thank the reviewer for their insightful comments and address each point/question in turn:
>
> **W1: R2D2 should be evaluated on more models.**
>
> **A1:** Unfortunately, Zephyr R2D2 is the only model available trained with R2D2 and training more base models with R2D2 is prohibitively expensive (see Table 1 in the paper and the overall response). We have thus focused on showing that our method is able to achieve a good robustness/utility trade-off and compared it where we could to R2D2, outperforming it.
>
> **W1.1: The authors should test their approach on more diverse safe answers** :
>
> **A1.1:** We explored generating diverse safety answers to increase the utility robustness trade-off in preliminary experiments but did not see any improvements. The common hypothesis is that a model is considered safe as long as the model provides any safe answer to a harmful query [Zou et al., 2023]. As diverse toward answers did not improve our results, we opted for the simpler approach. Using multiple away targets (harmful affirmative responses) for every query appeared to be more impactful, which is why we trained with 12 different possible away responses $\hat{y}$ for every given query. Note that the safety response is never used during evaluation and is just a training objective in the algorithm.
>
> **W1.2: Stronger attacks should be used for evaluation**:
>
> **A1.2:** We thank the reviewer for the remark. These attacks were not in the original submission because they are contemporary to our submission (NeurIPS guidelines suggest that any paper published on arXiv less than 2 months before the submission deadline should be considered contemporary work). However, we believe the reviewer suggestion to include these new attacks is a fantastic opportunity to showcase the width of the robustness of our model. We added results for the simple adaptive attack and an ICL attack for as many models as we could run during the rebuttal and will complete the remainder for the final paper. Both attacks are highly effective against the base models but cannot break the adversarial trained versions even when using $10000$ attack iterations and random restarts. We thank the reviewer for their suggestion, which we believe further strengthens our method’s empirical performance. Please see the attached PDF Table 2 and overall response for the results.
>
> **W1.3: Ablation studies on the choice of the preference optimisation algorithm should be conducted**:
>
> **A1.3:** As the reviewer points out, several preference optimization (PO) losses could be used in our adversarial training approach. We conducted preliminary experiments using DPO and found that IPO leads to less overfitting to the toward response (where the model rejects every request). A full ablation of all the preference losses feels to us a bit outside of the scope of this paper whose key contribution is to show that continuous robustness can robustify the models towards discrete attacks. However, we will surely add our results for DPO in the camera ready version.
>
> **W2: The proposed methods introduce hyperparameters that need to be tuned**:
>
> **A2:** It is true our method introduces a few more hyper-parameters (epsilon, attack iterations, beta), we believe tuning them is worth the improved robustness-utility trade-off. We acknowledge this limitation in the paper and also point out that the main baseline (R2D2) suffers from the same issue, with GCG having many more additional hyper-parameters to tune. Morever, it is common for adversarial training algorithms to require at least tuning of the epsilon and attack iteration parameters. Most methods, such as TRADES, additionally include a regularization parameter to control the robustness accuracy trade-off (beta in our case).
>
> Lastly, due to the cost of fine-tuning LLMs we were not able to conduct more than a few training runs for all models, which was sufficient to make the method work. Thus, we conclude that our approach is reasonably robust to hyperparameter choices.
>
> **W3: The presentation of the paper can be improved**
>
> **A3:** We thank the reviewer for their observation and added a description of \mathcal{L}' (the actual loss used for optimization after applying the cutoff) to the paper. We also reviewed every equation and made sure all the variables are defined.
>
> **Q1: Why do bigger models require smaller values of $\epsilon$?**
>
> **Q-A1:** We thank the reviewer for pointing this out and are happy to provide a hypothesis for this observation. We argue that larger models have been trained for longer, which leads to token embeddings that contain more information. Thus, smaller perturbations have a larger effect on the model. To investigate this theory, we conduct a continuous $\epsilon$-ball attack ($\epsilon = 0.05$) on the Phi and Gemma base models. Even though the initial robustness of the Phi model is larger, its robustness decreases substantially faster under attack. These findings seem to support our intuition. See the attached PDF for loss curves. We will add these experiments to the final paper.
>
> **Q2: Is it possible to add a regularization parameter to the adv. training algorithm like in TRADES?**
>
> **Q-A2:** Thanks for the nice question. Controlling the trade-off between robustness and utility is indeed crucial. For C-AdvIPO we ablate how robustness and utility depend on the $\beta$ parameter in Figure 3a of the paper. We demonstrate that small $\beta$ values increase robustness and decrease utility, which can be used to control the trade-off. Similarly, the trade-off can be controlled for C-AdvUL by choosing different weightings for the toward, away, and utility loss. Here a lower utility loss positively impacts the robustness while decreasing utility. We included the weighting parameters in Table 2 of Appendix A but did not clearly define them in the paper. We apologize for this oversight and will include the parameters in Equation (4) of the camera-ready version.

---

> > ### Comment · Reviewer_TJPv · 2024-08-13
> >
> > I thank the authors for detailed feedbacks. Most of my concerns have been address, so I am willing to improve my rating to 5. The revised manuscript should include stronger attacks (adaptive attacks) and more ablation studies. I agree with Reviewer Z2aE that the author should highlight the difference of their proposed defence method and existing ones.

---

> ### Author Response · Authors · 2024-08-13
>
> Thank you for your reply and feedback.
>
> We chose to evaluate our method with a subset of the strongest attack available at the point of submission (according to Harmbench [1]). Beyond the ICL attack and the adaptive attack that we performed for the rebuttal, we will conduct additional evaluations with the approaches proposed in [2, 3] for the final manuscript. These attacks have shown high ASR against robust models.
>
> We will add our ablation study on the behavior of $\epsilon$ for models of different sizes to the final manuscript. We are open to any further suggestions on ablation studies that would further improve our work.
>
> We will add a more profound discussion about the differences between previous approaches in encoder-decoder models and the presented adversarial training algorithm. More specifically, we will:
> - Elaborate on the different training settings, i.e., post-training stage for LLMs and pre-training in encoder-decoder models
> - Differences in the optimization target, i.e., robustness in classification settings vs. preference optimization or alignment finetuning
> - Differences in the optimization goal, i.e., improved generalization vs. adversarial robust alignment.
>
> We want to emphasize that robustness in classification problems differs substantially from adversarial robust alignment. In the classification setting, models are trained to be robust to minor input perturbations with respect to their prediction. In contrast, in the alignment setting, we are not concerned with the stability of predictions. Rather, we train the model to refrain from generating "toxic" outputs altogether. To achieve this, losses need to be adjusted for this task (such as combining different objectives or using preference optimization algorithms). None of the previous methods was designed for LLM alignment and they can not be employed for this task without further changes.
>
> Lastly, the majority of these approaches were used during the pre-training stage of encoder-decoder models. Applying these algorithms during the pre-training stage of an LLM exceeds our computational budget by orders of magnitude.
>
> [1] Zou et al., "HarmBench: A Standardized Evaluation Framework for Automated Red Teaming and Robust Refusal" (Feb. 2024)
>
> [2] Thompson T and Sklar M., "Fluent Student-Teacher Redteaming" (July 2024)
>
> [3] Liao et al., "AmpleGCG: Learning a Universal and Transferable Generative Model of Adversarial Suffixes for Jailbreaking Both Open and Closed LLMs" (May 2024)

---

### Official Review · Reviewer_Jk9f · 2024-07-08

**Soundness:** 3
**Presentation:** 3
**Contribution:** 3
**Rating:** 7
**Confidence:** 4

**Summary:**

This paper proposes adversarial training for LLMs, in which perturbations are created in the continuous embedding space rather than finding discrete suffixes. The proposed fast adversarial training algorithm (AdvUL) consists of two losses: the first strengthens the model against continuous embedding attacks computed on an adversarial behavior dataset, and the second ensures the final model's usefulness by fine-tuning utility data. Furthermore, the authors introduce C-AdvIPO, an adversarial variant of IPO that does not rely on utility data for adversarially robust alignment. The empirical evaluation of four models, Gemma, Phi3, Mistral, and Zephyr, at various scales, reveals that both algorithms improve LLM robustness against discrete attacks (GCG, AutoDAN, PAIR).

**Strengths:**

The Strengths of this paper include:

- The writing of this paper is clear, with neat formulas to fastly explain the proposed AT methods. Using the form of DPO/IPO in Eq. (5) seems reasonable to me.
- There are details to describe the experimental settings, and experiments are comprehensively done on different LLMs and datasets.
- The empirical improvements on the robustness against GCG, AutoDAN, and PAIR appear significant.

**Weaknesses:**

The Weaknesses of this paper include:

- The main difference between adversarial attacks/defenses on traditional models (e.g., CNNs) and those on LLMs is that there is no explicitly defined *threat model* when jailbreaking LLMs. Namely, the strategy to jailbreak an LLM could be quite flexible, and there is usually no constraint on human imperceptibility. AT is a strong defense under a given threat model (e.g., $8/255, \\ell\_{\\infty}$), but AT is observed to generalize badly to unseen attacks. While I'm experienced with AT in traditional settings, I'm still not very convinced that AT could be a good solution for LLMs.
- While GCG, AutoDAN, and PAIR are commonly evaluated attacks, they are relatively weak nowadays. The authors are encouraged to evaluate their models against more advanced and diverse attacks, such as ICL-based [1,2] and/or decomposing-based [3].
- To investigate the limits of C-AdvUL and C-AdvIPO, there should be a sanity check to directly perform attacks in the continuous embedding space.
- From my perspective, what C-AdvUL and C-AdvIPO do is connected to machine unlearning (i.e., unlearning the harmful knowledge), where [4,5] have used similar learning objectives. Their connections should be more discussed and empirically ablated.

Refs:\
[1] Many-Shot Jailbreaking\
[2] Improved Few-Shot Jailbreaking Can Circumvent Aligned Language Models and Their Defenses\
[3] A False Sense of Safety: Unsafe Information Leakage in 'Safe' AI Responses\
[4] Negating Negatives: Alignment without Human Positive Samples via Distributional Dispreference Optimization\
[5] Negative Preference Optimization: From Catastrophic Collapse to Effective Unlearning

**Questions:**

Why there are no experiments done on the Llama series of models? Is there any reason for this design choice?

**Limitations:**

Please see Weaknesses.

---

> ### Author Rebuttal · Authors · 2024-08-05
>
> We thank the reviewer for their insightful comments and address each point/question in turn:
>
> **W1: Is adversarial training suitable to robustify LLMs in the context of diverse threats?**
>
> **A1**: Thanks for initiating this discussion. We agree that the perturbations set in LLMs are much less constrained than in Computer Vision. However, both past work in computer vision [1, 2] and existing work in LLMs [3] indicate that variations of latent adversarial training can improve robustness against diverse threats. Our experimental results demonstrate that latent space adversarial training extrapolates robustness well beyond the continuous attack we train on, such as jailbreaking attempts and suffix attacks. Moreover, adversarial training is one of the only methods that stood the test of time and delivered reliable robustness improvements. In contrast, the majority of heuristic preprocessing approaches and other techniques were later broken. Overall, we don’t claim that AT can solve the problem entirely, but it presents a piece of the puzzle in robustifying the current generations of LLMs and is, therefore, worth studying. We are happy to engage in further discussions regarding this point.
>
> **W2: Could the authors include stronger attacks in their evaluation?**
>
> **A2:** Thanks for the references. These attacks were not in the original submission because they are contemporary to our submission (NeurIPS guidelines suggest that any paper published on arXiv less than 2 months before the submission deadline should be considered contemporary work). However, we believe the reviewer suggestion to include these new attacks is a fantastic opportunity to showcase the width of the robustness of our model. We agree that a thorough evaluation is crucial in this domain. We have added two attacks. One adaptive attack (as proposed by **TJPv**, which has shown very strong results on open source and proprietary models) and ICL. Our models increase the robustness against both attacks considerably compared to the base models, with most adversarially trained models achieving 100% robustness. See the PDF Table 2 and the overall response for the full results.
>
> **W3: A sanity check should be performed with continuous embedding space attacks to explore the limits of the robustness of the models**:
>
> **A3:** We agree with the reviewer that this is an important sanity check and have added this as an experiment. Against unbounded attacks, all models exhibit 0% robustness; against epsilon-ball attacks, the adversarial trained models show higher robustness than the base model (see also PDF and general comment).
>
> **W4: The connection of adversarial training to machine unlearning should be highlighted**:
>
> **A4:** We thank the reviewer for the great suggestion! Machine unlearning in the face of adversarial attacks is indeed closely related to the setting we consider here, we will amend the related work to highlight this. However, while the unlearning literature has proposed many new losses, such as NPO, they alone do not provide robustness to adversarial attacks (see the paper appendix B.2 Table 6 and attached PDF Table 1). We do believe that future work may look at how our adversarial training method may used in machine unlearning as well, but a full evaluation of this is beyond the scope of this paper.
>
> **Q1 Why are there no experiments done on the Llama series of models?**:
>
> **Q-A1**: No particular reason except that it’s utility performance is worse than the models we considered, but we have added LLama2 with our CAT method, which considerably increases the robustness of Llama2 (see the overall response and attached PDF Figure 2).
>
> (1) Casper et al., “Defending Against Unforeseen Failure Modes with Latent Adversarial Training” 2024
>
> (2) Laidlaw et al., "Perceptual Adversarial Robustness: Defense Against Unseen Threat Models.” ICLR, 2021
>
> (3) Dai et al., "Formulating robustness against unforeseen attacks."  NeurIPS, 2022

---

> > ### Comment · Reviewer_Jk9f · 2024-08-09
> >
> > I thank the authors for their responses and additional experiments, which addressed the majority of my concerns. I understand that stronger attacks like [1,2,3] occur concurrently with the submission, so I didn't take this as a negative point. Nonetheless, I encourage the authors to conduct more comprehensive evaluations in their paper revision, particularly against attacks that differ from the (implicit) assumptions of C-AdvUL and C-AdvIPO. In conclusion, I believe this paper provides a good defense against jailbreaks through extensive experiments. So I'd like to raise my score to 7.

---

> ### Author Response · Authors · 2024-08-10
>
> We thank the reviewer for their quick response and for increasing their score!
>
> We agree that further attack evaluations would improve our work and will look into adding decomposition attacks to our work. We welcome any further feedback to improve our paper and are thankful for your feedback.

---

### Author Rebuttal · Authors · 2024-08-05

We thank the reviewers for their efforts and suggestions. We included a pdf, which provides an overview of the new results. The following experiments have been added to the paper:

1. Training and evaluation of Llama2-C-AdvUL, which results in considerable robustness improvements.
2. Adaptive attack [1] and In-Context-Learning (ICL) attack. We demonstrate that our models are robust to both attacks. For the adaptive attack, we use the evaluation commands proposed in their GitHub repository and gpt-4-o as a judge.
3. Continuous attack sanity check. An unconstraint continuous attack breaks all our models. Adversarial trained models are more robust against $\epsilon$-ball attacks.
4. NPO baseline. We added two NPO and one additional IPO baseline without adversarial training. All of these models are not more robust to adversarial attacks than their base model counterparts.
5. R2D2 vs Ours Wall time. On an A100 a single step of the R2D2 took the 489.7 longer than a single step of C-AdvIPO. The complete training of RD2D would have taken 1991 times longer with the implementation provided in the official repository. As a result, we are unable to conduct R2D2 on other models.

**Open questions:** If we fail to address any remaining concerns, we will be happy to engage in more discussions.

(1) Andriushchenko, Maksym, et al. "Jailbreaking Leading Safety-Aligned LLMs with Simple Adaptive Attacks" (2024)

---

### Decision · Program_Chairs · 2024-09-25

**Decision:**

Accept (spotlight)

**Comment:**

**Summary of the Paper**

This paper proposes a new method for adversarial training based on the continuous embedding space of an LLM, rather than discrete optimizations. This method combines two losses, optimizing for attacks while preserving utility. The authors tested this method on various model families and scales, showing that the algorithms improved model robustness against discrete attacks while maintaining utility.

**Summary of Reviews**

- Reviewer Jk9f (Score 7 - Accept): The reviewer commends the clarity of the paper's presentation and the empirical data supporting the method's improvements on model robustness. They raise concerns about the defense's performance on unseen attacks, the paper's evaluation on potentially weak attacks, and further experiments investigating the limits of the two algorithms.
- Reviewer TJPv (Score 5 - Borderline Accept): The reviewer commends the paper's quality of writing and the performance improvements of the proposed method on adversarial training. They raise concerns over the comprehensiveness of experiments (e.g. only one safe response is matched for evaluations, and only relatively weak attacks are benchmarked) and the practicality of the method.
- Reviewer Z2aE (Score 5 - Borderline Accept): The reviewer commends the paper's overall presentation and the method's improvements in computational cost. They raise concerns over the comprehensiveness of experiments to support the method's performance improvement claim, the clarity of the technical descriptions of the method, the choice of models for benchmarking, and how the method compares to other defenses.

**Assessment**

Regarding reviewer Jk9f's concerns, the authors show that LLM robustness generalizes to out-of-distribution attacks, added additional metrics showing that the method improves robustness against newer attacks, added a sanity check to explore the limits of model robustness under this method, and added experimental data from Llama2.

Regarding reviewer TJPv's concerns, the authors explain their limitations in evaluating other more computationally expensive methods for comparison, explain that broadening safe answers did not improve model robustness, added results for newer attacks, and explain that other methods tend to have even less practicality in terms of hyperparameter tuning.

Regarding reviewer Z2aE's concerns, the authors provide metrics comparing the runtime differences between defenses, added Llama2 results, and explained that they only included defenses similar to their proposed method.

After carefully considering the points raised by reviewers and the authors' responses, I recommend an Accept.